# Studies of the Structure and Optical Properties of BaSrMgWO_6_ Thin Films Deposited by a Spin-Coating Method

**DOI:** 10.3390/nano12162756

**Published:** 2022-08-11

**Authors:** Luciana Punga, Abderrahman Abbassi, Mihaela Toma, Teodor Alupului, Corneliu Doroftei, Marius Dobromir, Daniel Timpu, Florica Doroftei, Laura Hrostea, George G. Rusu, Abdelati Razouk, Felicia Iacomi

**Affiliations:** 1Faculty of Physics, Alexandru Ioan Cuza University of Iasi, 11 Carol I Blvd., 700506 Iasi, Romania; 2LRPSI, Polydisciplinary Faculty of Beni-Mellal, Sultan Moulay Slimane University, Mghila BP 592, Beni-Mellal 23000, Morocco; 3CERNESIM-ICI, Alexandru Ioan Cuza University of Iasi, 11 Carol I Blvd., 700506 Iasi, Romania; 4Department of Exact and Natural Sciences, Institute of Interdisciplinary Research, Alexandru Ioan Cuza University of Iasi, 11 Carol I Blvd., 700506 Iasi, Romania; 5Petru Poni Institute of Macromolecular Chemistry, 1A Grigore Ghica Voda Alley, 700487 Iasi, Romania; 6RAMTECH, Institute of Interdisciplinary Research, Alexandru Ioan Cuza University of Iasi, 11 Carol I Blvd., 700506 Iasi, Romania; 7LGEM, FST, Sultan Moulay Slimane University, BP 523, Beni-Mellal 23000, Morocco

**Keywords:** thin films, double perovskite, tungstate, structure, optical properties

## Abstract

Highly transparent thin films with the chemical formula BaSrMgWO_6_ were deposited by spin coating using a solution of nitrates of Ba, Sr, and Mg and ammonium paratungstate in dimethylformamide with a Ba:Sr:Mg:W ratio = 1:1:1:1. XRD, SEM, EDX, and XPS investigations evidenced that annealing at 800 °C for 1 h results in an amorphous structure having a precipitate on its surface, and that supplementary annealing at 850 °C for 45 min forms a nanocrystalline structure and dissolves a portion of the precipitates. A textured double perovskite cubic structure (61.9%) was found, decorated with tetragonal and cubic impurity phases (12.7%), such as BaO_2_, SrO_2_, and MgO, and an under-stoichiometric phase (24.4%) with the chemical formula Ba_2−(x+y)_ Sr_x_Mg_y_WO_5_. From transmittance measurements, the values of the optical band gap were estimated for the amorphous (*E_gdir_* = 5.21 eV, *E_gind_* = 3.85 eV) and nanocrystalline (*E_gdir_* = 4.69 eV, *E_gind_* = 3.77 eV) phases. The presence of a lattice disorder was indicated by the high Urbach energy values and weak absorption tail energies. A decrease in their values was observed and attributed to the crystallization process, lattice strain diminution, and cation redistribution.

## 1. Introduction

Materials having a perovskite structure have been widely studied due to their numerous applications in many technological devices. Perovskites have a stoichiometry of ABX_3_, where the A cation (alkali or alkali earth metal, large with low charge) is 12 coordinated, the B cation (more electronegative with small radius) is 6 coordinated, and the X anion (oxygen or other anions) is coordinated by two B cations and four A cations [1,2,3,4,5]. 

The perovskite structure can be altered by substituting multiple cations at either the A or the B sites. Ordering of the octahedral-site cations and A-site cations in A_2_BB′X_6_ perovskites has been analyzed by many authors, who showed the alterations of the symmetry of both the undistorted aristotype and the distorted hettotypes, and that ordering of A-site cations is typical for anion-deficient perovskites [6,7,8,9]. 

Double perovskite structures can be derived from the perovskite structure when half of the B-site cations are replaced by another B′ cation, giving rise to the A_2_BB′X_6_ formula, where the A site can be occupied by Ca, Ba, Pb, Sr, and Na, and the B site, which is a six-fold coordinated transition metal ion or a light alkaline earth ion such as Mg, can be also occupied by Ti, Sn, W, Zr, Nb, and Ta [10].

Double perovskite structures are interesting due to their chemical flexibility, which offers the possibility to control their properties. Some recent works considered these materials as a new generation of materials that can have a significant impact on the evolution of material properties [11,12]. 

The structure and properties of double perovskite materials with either A_2_BB′O_6_ or AA’BB′O_6_ stoichiometry are dependent on cations’ distribution over the octahedral sites, the degree of cation inversion, and the size and electronic structure of transition metal cations B and B′ [13,14,15,16]. 

Some structural studies performed on BaSrMWO_6_ (M = Ni, Co, Mg) double perovskite oxides showed that the stable crystalline phase is the cubic phase with the space group Fm-3m [17]. Other studies performed on Ba_2-x_Sr_x_CoWO_6_ showed that, by increasing the Sr content, a phase transition occurs from cubic Fm-3m to tetragonal I4/m, and ultimately to monoclinic P21/n, and that at room temperature it is possible to have a mixture of phases [18]. It was suggested that A^2+^ ions have a significant effect on the formation of the A_2_ MgWO_6_ structure: a smaller ionic radius of A^2+^ favors a more distorted lattice, and a larger A^2+^ favors a cubic phase [19].

The stability and eventual distortion of the structure can be evaluated using Goldschmidt’s tolerance factor, which is useful in evaluating the types of oxides in the A_2−x_A′_x_BB′O_6_ double perovskite series [20,21]:(1)$$t=\frac{\left(1-\frac{x}{2}r_{A}\right)+\frac{x}{2}r_{A^{\prime }}+r_{O}}{\sqrt{2}\left(\frac{x}{2}r_{B}+\frac{x}{2}r_{B’}+r_{O}\right)}$$
where *r_A_*, *r_A′_, r_B_*, and *r_B′_* are the ionic radii of the A and B sites, and *r_O_* is the ionic radius of oxygen.

Depending on the value of *t*, the crystal structure may be diverse: for *t* > 1.05, the structure is hexagonal; for 1.05 > *t* > 1.00, it is cubic (Fm-3m); for 1.00 > *t* > 0.97, it is tetragonal (I4/m); and for *t* < 0.97, it is monoclinic (P21/n) or orthorhombic [21].

Several technologies are used for the synthesis of double perovskite structures as particles, in bulk, or as thin films (solid-state reaction, coprecipitation, pulsed laser deposition, rf sputtering, etc.) [17,18,19,20,21,22,23,24,25]. 

The double perovskites synthesized by solid-state reaction or coprecipitation methods, with the chemical formulas Ba_2_MgWO_6_ or Ba_2_BWO_6_ (B = Co, Ni, Zn), or other related systems, have shown great dielectric properties or high temperature sensitivity [22,24]. 

Until now, no study has been performed of thin films having the chemical formula BaSrMWO_6_ (M = Ni, Co, Mg), and previous structural studies have been performed on ceramics obtained by solid-state reaction [17,18,19,20,21,22,23]. 

Because these materials may have applications in devices for modern electronics, photovoltaics, and sensors, in this paper we present our preliminary results obtained for BaSrMgWO_6_ thin films deposited by spin coating on quartz and Si p substrates. 

In this investigation, we aimed to understand the structure and optical properties by undertaking a comparison with previous studies of bulk and actual thin films.

Before the experiment, a theoretical study of BaSrMgWO_6_ ceramic obtained by solid-state reaction, based on the DFT, was performed via the extraction of the band structure using WIEN2Kk Code to better understand the electronic properties [26].

## 2. Experimental Section

In a previous study, BaSrMgO_6_ samples were prepared using a solid-state reaction and the optical properties were investigated for the first time [26]. For this study, we intended to deposit thin films having the chemical formula BaSrMgO_6_. 

For thin film deposition, we selected a rapid and low-cost spin-coating technology that can be set up with relative ease [27]. The method allows the acquisition of uniform ultra-thin and thin films having a controlled thickness and a controlled chemical composition. The technology involves the preparation of preliminary solutions of nitrates of Ba, Sr, and Mg, and ammonium paratungstate in dimethylformamide, the preparation of the deposition solution by mixing the preliminary solutions to obtain a Ba:Sr:Mg:W ratio = 1:1:1:1; the usual substrate cleaning, the deposition process, and postdeposition annealings.

For thin-film deposition, we used a home-made spin coater at a speed of 3000 rot/min for 40 s for every layer. Quartz and (111) Si p substrates were used, and after every deposition, the samples were heated at 100 °C for 3 min and 200 °C for 5 min. The deposition process was repeated 10 times, and the samples were finally annealed at 800 °C for 1 h, resulting in sample BSMWO−I. Because sample BSMWO−I had an amorphous structure, the annealing process was repeated at 850 °C for 45 min, resulting in sample BSMWO-II (Table 1). 

The structure of the deposited thin films was analyzed by a Shimadzu LabX XRD-6000 Diffractometer with Cu Kα radiation (λ = 1.54059 Å) in Bragg–Brentano configuration. XRD patterns were recorded in the 2θ range of 10–80 degrees with a scan speed of 0.6 deg/min. A Verios G4 UC Scanning Electron Microscope (Thermo Scientific, Czech Republic, Brno) equipped with an energy dispersive spectrometer (EDS, EDAX Octane Elite) was used for thin-film morphology and elemental chemical composition. For this investigation, a Pt layer was deposited to prevent the electrostatic charge from accumulating on the sample surface. 

The information about the surface elemental chemical composition, and the chemical and electronic states of elements, was extracted from the XPS spectra registered with a SPHY-ULVAC VersaProbe 5000 device (AlKα, 1486.6 eV). The thin-film surfaces were contaminated by free carbon from the air; hence, the C 1s peak at 284.6 eV was used as a reference for all binding energies. The high-resolution XPS spectra of Ba 3d, Sr 3d, Mg 1s, W 4f, and O 1s were fitted with CasaXPS, using a Shirley background subtraction and mixed Gaussian–Lorentzian peak shapes.

The optical properties of thin films were examined in the wavelength range of 200 to 1200 nm by a double beam Shimadzu 2450 UV-Vis spectrophotometer, to obtain information about the band gap, Urbach energy, and weak absorption tail energy.

## 3. Results and Discussion

### 3.1. Structural Investigation

XRD patterns of thin films deposited on quartz are shown in Figure 1. For the BSMWO−I sample, XRD patterns evidence an amorphous structure, and, for the BSMWO−II sample, indicate a nanocrystalline structure containing the XRD peaks typical of the face-centered cubic structure of BaSrMgWO6 and some impurity XRD peaks, located at 2θ values of 26.70° and 27.49°, respectively (noted with * and **, respectively, in Figure 1). These peaks are generally attributed to impurity phases such as MeWO_4_ and Me_2_WO_5_ (Me = Ba, Sr, Mg). The presence of BaO_2_, SrO_2_, and MgO phases that may contribute to these XRD peaks may be caused by the formation of some under-stoichiometric phases [28,29,30,31,32,33].

Crystallographica Search-Match helped us to identify the XRD patterns of tetragonal BaO_2_ (F4/mmm, Pdf 3-1130), tetragonal SrO_2_ (P, pdf 3-872) XRD, and cubic MgO (Fm-3m, Pdf 75-1525), close to the XRD pattern of the BSMWO-−I thin film. Their XRD peaks may contribute to the XRD peaks observed at 2θ values of 26.70°, 27.49°, and 43.25°, respectively.

The XRD pattern of BSMWO−II was identified using Crystallographica Search-Match, and XRD peaks belonging to cubic phase Fm-3m were indexed according to PDF 01-080-3482 [16]. For the unit cell parameter of the cubic double perovskite phase, we obtained the value a = b = c = 8.1954 Ǻ, which is close to, but higher than, the value obtained previously for the bulk material (a = 8.0174 Ǻ) [17,23]. 

The crystallite size value was determined for all the XRD peaks observed for the double perovskite cubic phase with the Debye–Scherrer equation [34]:(2)$$D=\frac{0.9\cdot Å}{\beta cos\theta }$$
where *D* is the particle size, *λ* is the wavelength (1.5405 Å), *β* is the full width at half maximum, and *θ* is the Bragg angle. The crystallite sizes varied between 64.90 and 32.43 nm, and the median value, *D_m_*, was 42.70 nm (Table 1). This value is smaller than the values observed for the bulk double perovskite structures obtained by solid-state reaction, which may vary between 47 and 106 nm, or can be even larger, and are dependent on the heat treatment process and chemical composition [21].

The tolerance factor was calculated using Equation (1), with ionic radii values of 1.49, 1.32, 0.86, 0.74, and 1.35 Å, for Ba^2+^, Sr^2+^, Mg^2+^, W^6+^, and O^2−^, respectively, and was found to be 1.012 for the cubic crystalline sample, when x = 1.

The cubic structure is sustained by the lack of splitting of the (h00)-type and (hhh)-type reflections. 

Comparing the thin-film XRD pattern with the XRD patterns previously observed for the bulk BaSrMgO_6_ structure, obtained using the solid-state reaction method, we observe a preferential orientation of crystalline planes (220) parallel to the substrate and a shift to lower 2θ values [35]. The shift in XRD peaks attributed to (220), (331), and (400) crystalline planes to lower 2θ values, and in an XRD peak attributed to the (222) crystalline plane to higher 2θ values, may indicate a variation in cation distribution in A-type and B-type sites in comparison with the cubic structure of BaSrMgWO_6_ previously obtained by solid-state reaction [17]. Another factor that may make an important contribution to the shift in the XRD peaks is the uniform lattice strain [35]. The lattice strain (Table 1) was determined with the formula:(3)$$\varepsilon \;\left(\%\right)=\frac{a-a_{0}}{a_{0}}\cdot 100$$
where *a* is the unit cell parameter of the thin film and *a_0_* is the unit cell parameter of the bulk material having the same chemical composition [17]. The obtained positive value indicates a predominant relatively high tensile strain. The source of this tensile strain may be attributed to the thermal expansion mismatch between the thin film and the substrate during the annealing steps required in the formation of the double perovskite structure. It is known that in-plane tensile strain may induce changes in the oxidation state of cations, leading to the formation of oxygen vacancies, which are a source of instability [36].

### 3.2. Morphology and Elemental Chemical Composition

Figure 2a–d shows SEM images of BSMWO−I and BSMWO−II thin films deposited on quartz (Figure 2a,b) and on Si p (Figure 2c,d). The figures show the amorphous character of sample BSMWO−I, and the presence of cubic and tetragonal crystallites formed after the annealing at 850 °C in sample BSMWO−II.

Before the secondary annealing, BSMWO−I films exhibit a slightly granular structure (Figure 2a,c), which can be assigned to their columnar growth on substrates and spherical particles on the surface. Formation of precipitates on the thin-film surface was also observed in the literature [37].

Cracks are visible in the SEM image of sample BSMWO−I deposited on glass. They appeared due to the different thermal expansion coefficients of quartz and BSMWO thin film. No cracks were observed on the surface of BSMWO−I thin film deposited on (111) Si p, indicating smaller tensions in comparison to quartz substrate.

SEM images of BSMWO−II thin films deposited on quartz and (111) Si p (Figure 2b,d), that underwent supplementary annealing at 850 °C for 45 min, show that most of the globular precipitates become tetragonal or cubic and smaller in size, and exhibit a preferential orientation in the thin-film plane in agreement with the XRD pattern. This fact suggests that a portion of the precipitates was dissolved into the lattice, and that the tensions between the substrate and thin film were reduced (no cracks were observed for the quartz substrate).

Figure 3a,b shows the EDX spectra belonging to different areas of BSMWO−II, and confirms that the sample contains all elements of the raw material inputs in comparable contents. Table 2 presents the atomic elemental compositions for the investigated area.

By extracting the quartz substrate chemical composition and the surface adventitious carbon, it was possible to determine a chemical formula for the thin film and the surface precipitates. EDX analysis evidenced a possible stoichiometric formula for the double perovskite BaSrMgWO_6_, and possible stoichiometric formulas for surface oxides such as BaO_2_, SrO_2_, and MgO [37,38].

These oxides indicate the presence of other possible phases that may result during the deposition and annealing processes and are suggested by the XRD pattern and the high value of tensile strain.

Because SEM analysis sustains the W:Ba:Sr:Mg ratio = 1:1:1:1, we suppose that a certain percentage of oxide phases is the result of the migration of some cations outside the stoichiometric lattice leaving an under-stoichiometric lattice, according to these two possible chemical relations:

BaSrMgWO_6_ →Ba_2−(x+y)_ Sr_x_Mg_y_WO_5_ + Ba_x+y_O+ Sr_2−x_O + Mg_2−y_O(4)
where 0 ≤ x, y ≤ 2; and:
BaSrMgWO_6_ →Ba_1−(x+y)_ Sr_x_Mg_y_ WO_4_ + Ba_x+y_O + Sr_1−x_O + Mg_1−y_O(5)
where 0 ≤ x, y ≤ 1.

The formation of peroxide phases under the conditions we used for thin-film deposition and annealing is also possible because BaO and SrO can easily adsorb oxygen from the air to form BaO_2_ and SrO_2_.

### 3.3. Chemical States

XPS surface analysis was focused on determining whether Ba, Sr, and Mg can be found in the stoichiometric or under-stoichiometric double perovskite lattice and in the surface oxides, since the registered binding energy of each given element’s electron orbital provides information about its specific chemical state. 

Although EDX spectra showed a content of W similar to that of Ba, Sr, and Mg, the high-resolution XPS spectrum of W4f (Figure 4a) has a small intensity. This situation is caused by the surface species, which are also evidenced by SEM images. The asymmetric broad peak, located at a binding energy higher than that observed for W^4+^ species, indicates inhomogeneities in the chemical composition. The presence of superficial oxides indicates the possible existence of W^5+^. This chemical state can be the result of the tensile strain that induces changes in the oxidation state of tungsten, leading to the formation of oxygen vacancies, the migration of Ba, Sr, and Mg cations to the surface, and the formation of an under-stoichiometric lattice. 

It was possible to deconvolute the XPS W4f spectrum into two doublets with equal FWHM (2.5 eV) with a 4f_7/2_–4f_5/2_ doublet separation of 2.2 eV and an area ratio of 4:3. The peak positions located at 37.60 and 39.73 eV, respectively attributed to 4f_5/2_ and 4f_7/2_, indicate the W^6+^ state, typical for the W-O bond in the cubic double perovskite structure [39,40,41]. The doublet with the XPS peaks located at 36.53 and 38.73 eV may be attributed to the W^5+^ state. The XPS peak area of W^5+^ represents 38.1% of the total spectrum.

For the deconvolution of the high-resolution core level Ba 3d, Sr 3d, and Mg 1s spectra, we considered Relation (4), the formation of BaO_2_, SrO_2_, and MgO compounds, and the states in an under-stoichiometric structure (evidenced by the presence of W^5+^). For an appropriate fit of Ba 3d, Sr 3d, and Mg 1s spectra, we included three different species in the fitting routine. We considered the high and low binding energy species as surface and bulk species, respectively. 

The spin–orbit splitting energy of the Ba-3d state was approximately 15.54 eV for each of the three species we introduced into the fitting routine, with a d_5/2_ to d_3/2_ intensity ratio of 3:2. The doublets attributed to different species were separated by approximately 1.8 eV (Figure 4a) and 2.25 eV, and had the same full width half maximum (FWHM) of 2.2 eV. 

Since barium is a highly electropositive element, in barium oxide compounds the state is always 2^+^ and the chemical shift is a result of a local potential change caused by different anion coordination. As the coordination of negative ions is higher, the local electron potential increases and the binding energy decreases [42]. 

In the stoichiometric perovskite structure, Ba is 12-fold, and, in the under-stoichiometric and peroxide structures, it is 10-fold; therefore, the binding energy of Ba^2+^ in the cubic double perovskite structure will be lower than that of Ba^2+^ in the under-stoichiometric Ba_2−(x+y)_ Sr_x_Mg_y_ WO_5_ structure or in tetragonal BaO_2_ [28,37,43]. The attribution of the peaks with higher binding energy to BaO_2_ agrees with the results obtained by other authors [37].

The best fits in the deconvolution of Ba 3d XPS were obtained, of 61.9% for Ba^2+^ species in BaSrMgWO_6_, 25.4% for Ba^2+^ in under-stoichiometric phases of type Ba_2−(x+y)_ Sr_x_Mg_y_ WO_5_, and 12.7% for surfaces oxides, as expected (Table 3) [44,45,46]. 

The Sr 3d XPS high-resolution spectrum was fitted with three doublets, restricted by an equal FWHM of 2 eV; a fixed doublet separation of 1.7 eV (spin orbit splitting); and a ratio of the area of d_5/2_ to d_3/2_ of 3:2 [26,46]. Based on the best fit, the following attributions for the resulted XPS 3d_5/2_ peaks were made: 133.14 eV is typical of Sr^2+^ species located in the cubic double perovskite; 135.12 eV is characteristic of Sr^2+^in a Ba_2−(x+y)_ Sr_x_Mg_y_ WO_5_ under-stoichiometric lattice; and 135.33 eV is characteristic of Sr^2+^ in the tetragonal lattice of SrO_2_ (Figure 4b, Table 3) [47,48,49]. The two peaks belonging to different species are separated by 2.0–2.2 eV. The repartition of different Sr^2+^ species resulting from the best fit was the same as the repartition of Ba^2+^ species.

Like barium and strontium, magnesium is a highly electropositive element, and in oxide compounds its state is Mg^2+^. For the deconvolution of the high-resolution XPS Mg 1s spectrum, we used the same repartition of Mg^2=^ species as we previously used for Ba^2+^ and Sr^2+^. The best fit is shown in Figure 4d and the deconvolution results are indicated in Table 3. The obtained results are in good agreement with the literature [37,38,49,50]. The species were fitted with peaks of equal FWHM (2.2 eV), and an energy separation between peaks of 1.6 and 2.8 eV. 

For the XPS O1s spectrum, it was necessary to introduce five peaks with different FWHM values into the fitting routine (Figure 4e, Table 3). The peak with the lowest binding energy of 530.11 eV (2.2 eV) is characteristic of the stoichiometric double perovskite lattice, and the peak with the binding energy of 531.87 eV (2.0 eV) is characteristic of surface oxide species. The peak with the binding energy of 531.63 eV is typical for an under-stoichiometric lattice. The other two peaks, located at 532.7 eV (1.89 eV) and 533.7 eV (2.39 eV), are characteristic of species adsorbed on the surface, such as free OH groups, and OH in free water molecules and oxygen molecules, O_2_. These peaks can be also related to XPS peaks located at 286.27 and 287.42 eV in the high-resolution C 1s spectrum [51,52,53,54]. 

However, on perovskite surfaces, adsorbed oxygen species play a role [51,53]. We consider that surface SrO_2_ and BaO_2_ species result in a separate oxygen surface species in conjunction with hydroxylation. 

The area of XPS peaks, *I_j_*, attributed to different elements present in the three species identified as BaSrMgWO_6_, Ba_2−(x+y)_ Sr_x_Mg_y_WO_5_, and surface oxides (the adsorbed phases and adventitious carbon were neglected) were used to determine the thin-film chemical elemental composition, in at%, using the sensitivity factors, *S_j_* (O 1s = 0.733, Sr 3d = 1.992, Ba 3d = 7.343, W 4f = 3.863, Mg 1s = 1.035) and the relation [55]:(6)$$X_{p}\left(\%at\right)=\frac{I_{p}/S_{p}}{\sum_{j}\frac{I_{j}}{S_{j}}}$$
where *X_p_* is the atomic concentration, in percent, of a certain element. The obtained results are presented in Table 4.

By correlating XRD, SEM, EDX, and XPS results, it was possible to identify the phases formed in the thin films and obtain information about the processes that take place during annealing. 

### 3.4. Optical Properties

At the interaction of a material with photons, a fundamental absorption may occur due to excitonic processes or band-to-band transitions [56,57]. The band-to-band transition at the fundamental edge may be direct or indirect. Previous theoretical and experimental studies performed on bulk BaSrMgWO_6_ evidenced a direct band-to-band transition and an optical bang gap energy, *E_g_*, of 3.7 eV [26]. Another recent theoretical study performed on this compound established an indirect band gap with an optical band gap energy of 3.11 or 3.03 eV when the local density approach (LDA) or the generalized gradient approximation (GGA) was used [56].

Figure 5a–e shows the results obtained from transmittance measurements.

The thin-film transmittance was higher than 80% in the UV range and exceeds 90% in the visible range (Figure 5a). The thin film annealing process at 850 °C determined the crystallization process and, consequently, the increase in the absorption coefficient in UV and a red shift of the transmittance spectrum. For BSMWO−II thin film, two absorption bands became visible at around 350 and 450 nm, which generally are attributed to the charge transfer of W^6+^-O^2−^ and distorted (WO_6_)^6−^ octahedrons [39]. Both transmittance spectra are blue shifted compared with the bulk transmittance spectrum [26].

The absorption coefficient was determined by using the relation generally used for transparent thin films:(7)$$\alpha =\frac{1}{d}ln\left(\frac{100}{T}\right)$$
where *d* is the thin-film thickness in cm, and *T* is the transmittance in % [57,58].

Plots of the absorption coefficient versus wavelength are shown in Figure 5b. The absorption coefficients increase sharply with the increase in photon energy. The photon energy dependence of the absorption coefficient of sample BSMWO−II shows higher values compared with the absorption coefficient of sample BSMWO−I until 1.9 eV (650 nm), when its values become comparable and even smaller at lower energy values (3177.4 cm^−1^ for BSMWO−II and 4409.5 cm^−1^ for BSMWO−I at 1.55 eV). Both plots evidence a tail that can be attributed to the presence of disorder and defects, and their shape enables both direct and indirect band-to-band transitions due to the more disordered layer formed on the surface.

For the optical band gap energy determination, we used the Tauc formula [59,60]:(8)$$h\nu \alpha =A{\left(h\nu -E_{g}\right)}^{n}$$
where *A* is an energy-dependent constant, *E_g_* is the optical band gap energy, *h* is the Planck constant, *ν* is the photon frequency, and *α* is the absorption coefficient. The exponent *n* has the value *n* = 1/2 for a band-to-band direct transition and the value *n* = 2 for an indirect transition. Figure 5c,d shows the plots of (ℎν𝛼)^2^ versus *hν* for the direct transition and the plots (ℎν𝛼)^1/2^ versus *hν* for the indirect transition. 

The values obtained for the optical band gap energies (Table 5) are higher than the value obtained for the bulk material, both for the direct transition (*E_gdir_* = 3.70 eV) and indirect transition (*E_gind_* = 3.11 eV) [26,55]. These differences can be attributed to the fact that, for the double perovskite thin films having a thickness of 150 nm, the optical band gap energy is predominantly influenced by the quantum size effect.

During band-to-band transitions due to disorder, the density of states in the valence and conduction band tails off into the energy gap and, consequently, the absorption coefficient *α*(*E*) also tails off in an exponential manner. The energy associated with this tail is referred as Urbach energy, and can be calculated by the following equation [57]:(9)$$\alpha \left(h\nu \right)=exp\left(\frac{h\nu }{E_{U}}\right)$$

There is also a weak absorption tail for those transitions taking place from one localized tail state above the valence band to another localized tail state below the conduction band, in the weak absorption region. The energy associated with this tail is referred to as *E_T_* and can be calculated with a relation such as Equation (9). If Urbach energy is caused by disorder, the weak absorption tail energy is caused by defects and lattice strain.

Figure 5e shows the plots *lnα* = *f*(*hν*) that allow the determination of *E_U_* and *E_T_* by fitting the linear portion of the curve with a straight line in the Urbach and weak absorption tail regions; the reciprocal of the slope yields the values of *Eu* and *E_T_*, respectively (Table 5). 

The high values of Urbach energy and weak absorption tail energy indicate disorder, in agreement with SEM, EDX, and XPS results. The decrease in Urbach and weak absorption tail energy with thin film annealing at higher temperatures is due to the improvement in the crystallinity of the films and elimination of a portion of the defects and lattice strain. 

## 4. Conclusions

Using a low-cost spin-coating technology, we successfully grew highly transparent thin films with a cubic Fm-3m double perovskite structure, having a thickness of 150 nm and the chemical formula BaSrMgWO_6_.

XRD, SEM, EDX, and XPS investigations evidenced that annealing at 800 °C for 1 h resulted in an amorphous structure having precipitate on its surface. Supplementary annealing at 850 °C for 45 min succeeded in starting the crystallization process and dissolving a portion of the precipitates. Results showed a textured cubic structure (61.9%) and the presence of some tetragonal and cubic impurity phases (12.7%) such as BaO_2_, SrO_2_, and MgO, and an under-stoichiometric phase (24.4%) having the chemical formula Ba_2−(x+y)_ Sr_x_Mg_y_WO_5_.

It also possible to conclude, considering the percent of phases as determined from XPS analysis, that the result of our experiments is a thin film having a cubic stoichiometric double perovskite structure with a thickness of 93 nm, decorated with an under-stoichiometric layer of 38 nm and a layer of oxide nanoparticles with a thickness of 19 nm (Figure 5f). The under-stoichiometric layer and the oxide particles play an important role in oxygen surface species in conjunction with hydroxylation.

Investigation of optical properties enabled determination of the direct and indirect optical band gap energies and provided information about Urbach and weak absorption tail energies. The values obtained for the amorphous and nanocrystalline thin films are higher compared with the values observed in the previous studies of the bulk material, which is due to the quantum size effect and the content of defects. The indirect optical band gap can be attributed to the formation of the under-stoichiometric layer because of cation migration on the surface and the formation of oxide precipitates.

Considering the interesting properties that the surface peroxide, oxide nanoparticles, and under-stoichiometric phase may create in the thin film, future investigations on sensor, photocatalytic, and dielectric properties will be performed. Furthermore, it will be interesting to study the effect of annealing performed at a higher temperature on the structure and thin-film optical properties.

## Figures and Tables

**Figure 1 nanomaterials-12-02756-f001:**
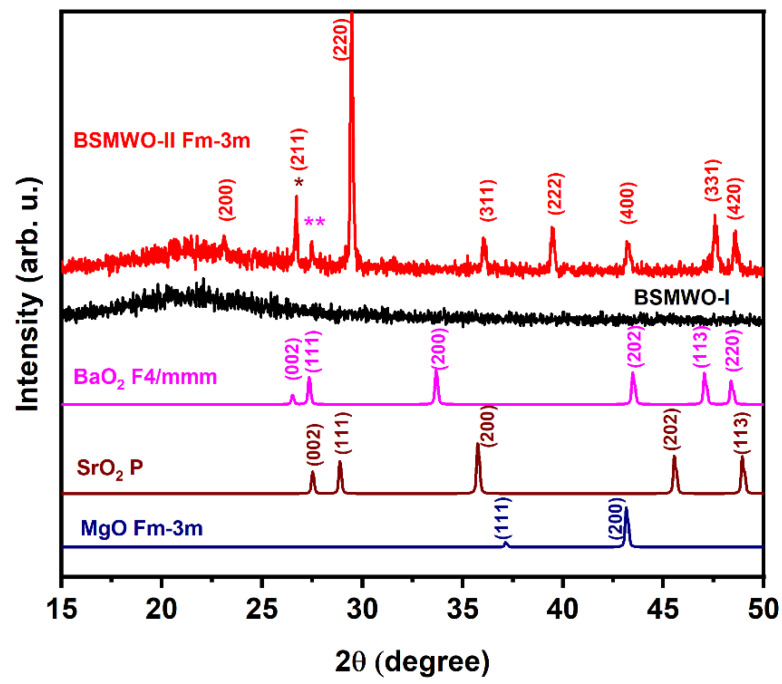
XRD patterns of thin films deposited by spin coating and annealed at different temperatures: BSMWO−I—black line, 800 °C, 1 h; BSMWO−I—red line, 800 °C, 1 h and 850 °C, 45 min. XRD patterns are compared with three possible impurity phases that may exist. The XRD peaks noted with * and ** may have contributions from the tetragonal BaO_2_ and SrO_2_ phases.

**Figure 2 nanomaterials-12-02756-f002:**
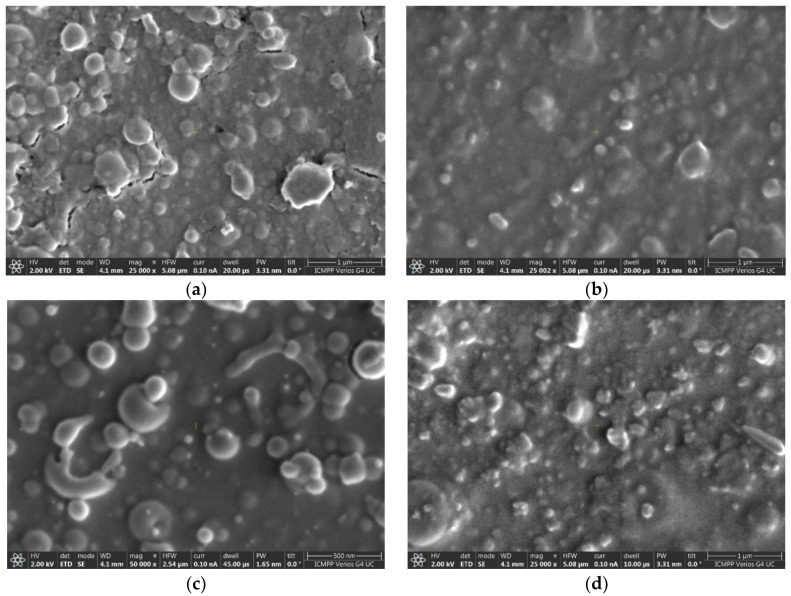
SEM images: (**a**) BSMWO−on quartz substrate; (**b**) BSMWO−II on quartz substrate; (**c**) BSMWO−I on (111) Si p substrate; (**d**) BSMWO−II on (111) Si p substrate.

**Figure 3 nanomaterials-12-02756-f003:**
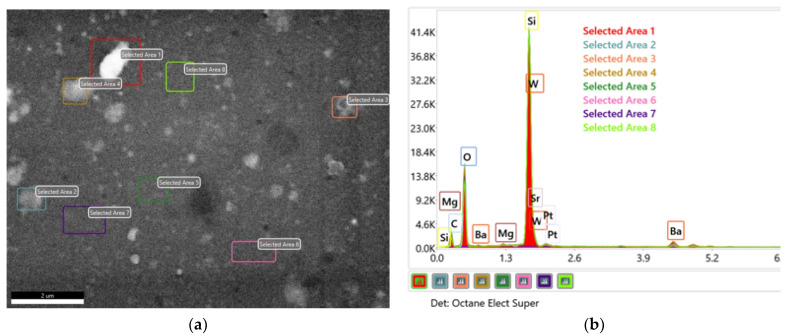
EDX spectra belonging to 8 different areas of the BSMWO−II SEM image: (**a**) SEM image; (**b**) EDX spectra. The investigated area and the corresponding spectra are marked with the same colors.

**Figure 4 nanomaterials-12-02756-f004:**
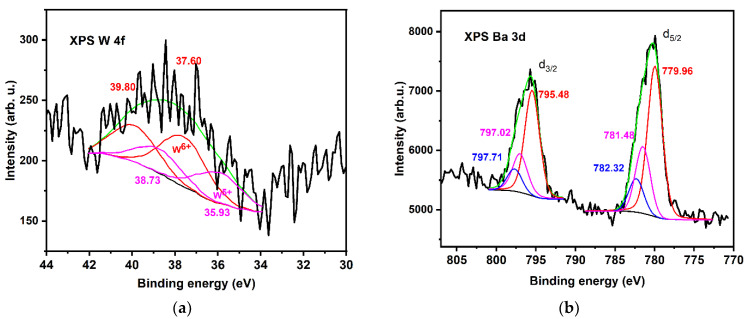
High-resolution XPS spectra of BSMWO−II thin film: (**a**) Ba 3d; (**b**) Sr 3d; (**c**) Mg 1s; (**d**) W 4f; (**e**) O 1s; (**f**) C 1s. Red line indicates species in cubic coordination in double perovskite structure, blue line indicates surface species, and orange line indicates adsorbed species.

**Figure 5 nanomaterials-12-02756-f005:**
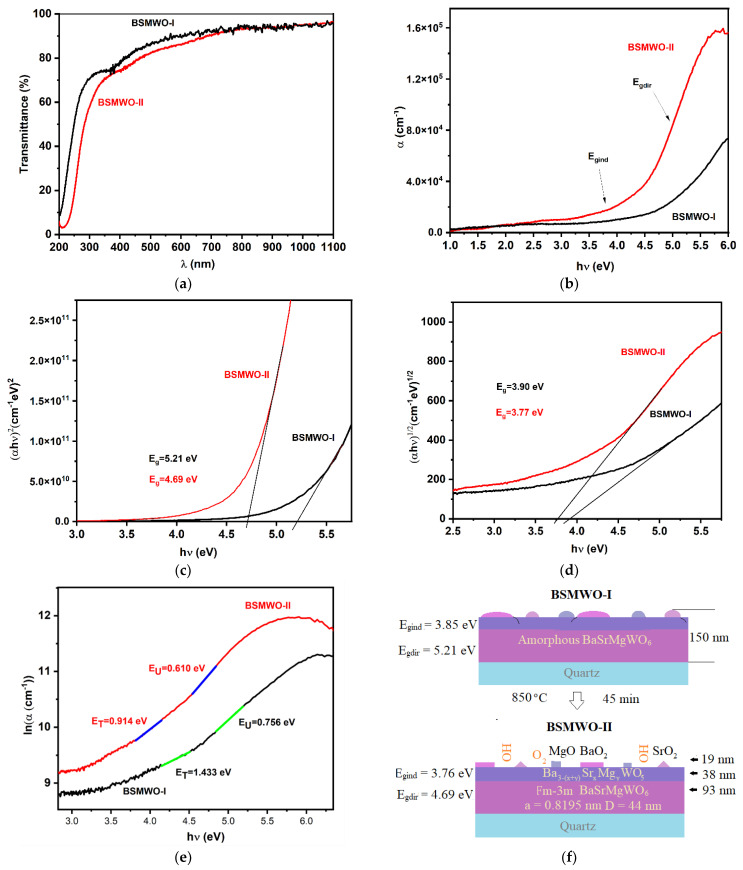
Optical properties of BSMWO−I and BSMWO−II thin films: (**a**) transmittance spectra; (**b**) absorption coefficient; (**c**) determination of optical band gap energy for a direct transition; (**d**) determination of optical band gap energy for an indirect transition; (**e**) determination of Urbach and UTW energies; (**f**) thin-film characteristics.

**Table 1 nanomaterials-12-02756-t001:** Structural characterization of thin films.

Sample	Post Deposition Annealing	a Å	V Å^3^	D_m_ nm	Symmetry	Tolerance Factor, t	ε %
BSMWO−I	800 °C, 1 h	-	-		amorphous		
BSMWO−II	800 °C 1 h and 850 °C, 45 min	8.1954 ± 0.0757	550.436 ± 15.257	42.70 ± 0.05	Fm-3m	1.012 ± 0.005	2.22 ± 0.05

**Table 2 nanomaterials-12-02756-t002:** EDX chemical elemental composition of BSMWO−II thin film.

Elements	Area 1 Atomic %	Area 2 Atomic %	Area 3 Atomic %	Area 4 Atomic %	Area 5 Atomic %	Area 6 Atomic %	Area 7 Atomic %	Area 8 Atomic %
C K	34.6	31.1	30.4	34.1	28.5	27.8	28.7	30
O K	44.1	46.6	47	44.7	47.6	47.3	47.5	46.3
Si K	19.6	20.9	21.2	19.3	22.7	22.8	22.6	22.5
Mg K	0.2	0.3	0.3	0.3	0.3	0.3	0.3	0.3
Sr L	0.3	0.3	0.3	0.3	0.2	0.4	0.4	0.4
Ba L	0.8	0.5	0.4	0.8	0.2	0.2	0.2	0.2
W L	0.2	0.1	0.2	0.2	0.2	0.2	0.2	0.2
Pt L	0.2	0.2	0.2	0.2	0.2	0.1	0.2	0.2

**Table 3 nanomaterials-12-02756-t003:** Binding energies of elements Ba 3d, Sr 3d, Mg 1s, O 1s, and C 1s observed in BSMWO−II high-resolution spectra, (eV), peak area (%), and attribution.

Compound	BE (eV)								
	Ba 3d		Sr 3d		Mg 1s	W 4f		O 1s	C1s
	3d_5/2_	3d_3/2_	3d_5/2_	3d_3/2_	1s	4f_7/2_	4f_5/2_	1s	1s
BaSrMgWO_6_	779.96	795.48	133.18	134.88	1304.27	37.60	39.90	530.11	
Peak area %	61.9		61.9		61.9	61.9		33.3	
Ba_2−(x+y)_ Sr_x_Mg_y_WO_5_	781.48	797.02	135.12	136.73	1305.33	35.93	38.73	531.63	
Peak area, %	25.4		25.4		25.4	38.1		22.7	
BaO_2_, SrO_2_ MgO	782.32	797.71	136.33	137.33	1306.33			531.87	
Peak area, %	12.7		12.7		12.7			11.4	
C-C									284.6
Peak area, %									46.7
OH; C-OH								532.68	286.27
Peak area, %								18.4	31.6
H_2_O, O_2_ C=O								534.00	287.41
Peak area, %								14.2	21.7

**Table 4 nanomaterials-12-02756-t004:** Elemental chemical composition of BSMWO−II thin film.

Compound	Elements, Atomic %				
	O	Ba	Sr	Mg	W
BaS rMgWO_6_	40.39	5.38	5.38	5.38	5.38
Ba_2−(x+y)_ Sr_x_Mg_y_WO_5_	16.57	2.21	2.21	2.21	3.31
BaO_2,_ SrO_2_, MgO	8.29	1.10	1.1	1.10	0.00
Total	65.25	8.69	8.69	8.69	8.69

**Table 5 nanomaterials-12-02756-t005:** Thin-film optical properties: transmittance, *T*; absorption coefficient, α; bang gap energy for direct and indirect transitions, *E_gdir_*, *E_gind_*: Urbach energy, *E_U_*; and weak absorption tail energy, *E_T._*

Sample	*d* nm	*T* (450 nm) %	α (2.75 eV) cm^−1^	*E_gdir_* eV	*E_gind_* eV	*E_U_* eV	*E_T_* eV
BSMWO−I	150	88.3	6734.6	5.21	3.90	0.756	1.433
BSMWO−II	150	83.3	9705.7	4.69	3.77	0.610	0.914

## Data Availability

Not applicable.

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
