# Peer review of "Studies of the Structure and Optical Properties of BaSrMgWO6 Thin Films Deposited by a Spin-Coating Method"

_nanomaterials, 2022, doi:10.3390/nano12162756_

Round 1

Reviewer 1 Report

The manuscript is presenting an interesting topic on developing perovskite materials on BaSrMgWO6 ceramic obtained by solid-state reaction, based on  DFT simulations. The few comments and questions are due before we proceed with this submission:

1) crystallite sizes varied between 77.4 and 32.43nm but the median value was 44.90nm, is this in the same range as in the literature? please mention and compare with the literature results. if not correlating with literature, please mention why?

2) The shift is observed in XRD peaks (220), (311) to lower 2theta values indicating a variation in cation 160 distribution in A type and B type sites. is this correlating with XPS measurements or other cation measurements or you only could compare that with the literature?

3)the XPS and SEM analysis must be cited to https://doi.org/10.3390/coatings12060823

4)precipitates on the thin film surface were observed in SEM images. please explain the source of these precipitates.

5) the oxide phase formation is really interesting part of this manuscript. this was sourced to migration of cations outside the stoichio. lattice. do you confirm this by XPS or XRD?

when authors take my comments in and reply to my questions, i can reconsider my decision. all comments must be considered and not being ignored in the reply.

Author Response

Response to Reviewer 1 Comments

Thank you for your thorough review, those comments are all valuable and very helpful for revising and improving our paper.

Point 1: crystallite sizes varied between 77.4 and 32.43nm but the median value was 44.90nm, is this in the same range as in the literature? please mention and compare with the literature results. if not correlating with literature, please mention why?

Response 1: This is the first study on depositing a BaSrMgWO6 thin film, all the other studies were performed on bulk ceramics and authors gived no information about grain sizes, so we could not make a comparation with other studies. But, generally the crystallite sizes are larger for the bulk phases obtained by conventional solid-state technique, so we introduced a new phrase in 3.1. Structural investigation:

“This value is smaller compared to the value observed for the bulk double perovskite structure obtained by solid-state reaction, that may vary between 47nm and 106 nm or can be even larger, being dependent on the heat treatment process and chemical composition [21].”

[21] Alsabah, Y. A.; AlSalhi, M. S.; Mustafa, E. M.; Elbadawi, A. A.; Devanesan, S.; Siddig, M. A.; Synthesis, Phase Transition, and Optical Studies of Ba2−xSrxZnWO6 (x = 1.00, 1.25, 1.50, 1.75, 2.00) Tungsten Double Perovskite Oxides, Crystals 2020, 10, 299. https://doi.org/10.3390/cryst10040299

Point 2: The shift is observed in XRD peaks (220), (311) to lower 2theta values indicating a variation in cation 160 distribution in A type and B type sites. is this correlating with XPS measurements or other cation measurements or you only could compare that with the literature?

Response 2: There was no similar situation in the literature for this compound. But generally the shift of XRD peaks can be a consequencce of a change in cation distribution or of a lattice strain.

We introducednin text the:

“Another factor that may have an important contribution to the XRD peaks shift lo lower 2q values and to the increase in the unit cell parameter is the uniform lattice strain [35]. The lattice strain (Table 1) was determined with the formula:

                                    (3)

where a is the unit cell parameter of thin film and a0 is the unit cell parameter of the bulk with the same chemical composition [17]. The obtained positive value indicates a predominant relatively high tensile strain. The source of this tensile strain may be attributed to the thermal expansion mismatch between the thin film and substrate during the annealing steps required in the formation of double perovskite structure. It is known that in-plane tensile strain may induce changes in oxidation state of cations leading to the formation of oxygen vacancies, being a source of instability [36].”

[36] Chong, C.; Liu, H.; Wang, S.; Yang, K. First-Principles Study on the Effect of Strain on Single-Layer Molybdenum Disulfide. Nanomaterials 2021, 11, 3127. https://doi.org/10.3390/nano11113127

We also evidenced by XPS the presence of an under-stoichimetric phase and of some oxides formed on the surface (also evidenced by SEM), so we supposed that the XRD shift could be caused also by a cation distribution different from the distribution observed for the bulk.

We introduced in the text at XPS analyses:

“The presence of superficial oxides indicates the possible existence W5+. This chemical state can be a result of tensile strain that induced changes in the oxidation state of tungsten leading to the formation of oxygen vacancies, to the migration of Ba, Sr and Mg cations to the surface and to the formation of an under-stoichiometric lattice.”

Point 3: the XPS and SEM analysis must be cited to https://doi.org/10.3390/coatings12060823

Responce 3: We introduced at XPS and SEM analyses doi.org addresses where were available.

We observed that the reference [38] was the same with reference [35], so was deleted and the reference numbers were changed acordingly.

Point 4: precipitates on the thin film surface were observed in SEM images. please explain the source of these precipitates.

Response 4: We explained at point 2 and in the text: The migration of cations to the thin film surface followed by the formation of an under-stoichiometric structure and oxides formation is caused primordially by the temarcable value of tensile strain caused by the thermal expansion mismatch between the thin film and substrate. We also mentioned in the text:

“These oxides indicate the presence of other possible phases that may result during the deposition and annealing processes and are suggested by XRD pattern and by the high value of tensile strain.

Because SEM analysis sustains the ratio W:Ba:Sr:Mg=1:1:1:1 we suppose that a certain percent of oxide phases are the result of the migration of some cations outside the stoichiometric lattice, living an under-stoichiometric lattice according to these two possible chemical relations:

BaSrMgWO6 ® Ba2-(x+y) SrxMgyWO5 + Bax+yO + Sr2-xO + Mg2-yO                        (4)

where 0 £ x £ 2,

BaSrMgWO6 ® Ba1-(x+y) SrxMgyWO4 + Bax+yO + Sr1-xO +Mg1-yO                         (5)

where 0£x, y £1.

The formation of peroxide phases in the conditions we used for thin film deposition and annealing is also possible. since BaO and SrO can easily adsorb oxygen from the air to form BaO2 and SrO2.”

XPS analysis proves the formation of oxide phases and of under-stoichiometric phase according to .relation (4), but considering also the formation of peroxide phases – sustained by SEM and XRD.

We corrected relation (5). In the original text was uncorrectly written, as below:

“BaSrMgWO6 ®Ba1-(x+y) SrxMgyWO5+Bax+yO+ Sr1-x +Mg1-yO            ”

Because we intoduced befor a new relation, we changed the relation’s number acordingly.

Point 5: the oxide phase formation is really interesting part of this manuscript. this was sourced to migration of cations outside the stoichio. lattice. do you confirm this by XPS or XRD?

Response 5: The deconvolution of XPS spectra confirmed the formation of BaO2, SrO2 and MgO and also the formation of the under-stoichimetric phase of type Ba2-(x+y) SrxMgyWO5. These phases have also contrubutions to XRD pattern as was mentioned in Fig. 1. EDX analysis also confirmed their formation.

Reviewer 2 Report

The authors presented a material BaSrMgWO6 innovation by using spin coating method. These materials may have applications in devices for modern electronics, photovoltaics, and sensors.  The preliminary results obtained for BaSrMgWO6 thin films deposited by spin coating on quartz and Si p substrates.  Authors understand the structure and optical properties making a comparison study between previous studies on bulk and actual thin films of BaSrMgWO6. The presented results are interesting and significant. This manuscript is recommended to be published after minor revision. The suggested changes are as follows:

1.     What kind of technological aspects are consider while making this spin coated BaSrMnWO6 thin films?

2.     Which class of technology with the new materials are applicable?

3.     This is great work. What are the technology authors are intended to mention as a future materials?

4.     What are the advantages to deposit the BaSrMnWO6 with spin coating?

Author Response

Response to Reviewer 2 Comments

Thank you for your comments and questions, they are helpful for manuscript improvement.

Point 1: What kind of technological aspects are consider while making this spin coated BaSrMnWO6 thin films?

Response 1:

We added in the text at Experimental Sectiion: “For thin film deposition we selected a low-cost spin coating technology that is a rapid method with a relative ease in setting up the process [25]. The method allows to obtain uniform ultra-thin and thin films with a controlled thickness and a controlled chemical composition. The technology involves the preparation of preliminary solutions of nitrates of Ba, Sr and Mg and ammonium paratungstate in dimethylformamide, the preparation of deposition solution by mixing the preliminary solutions to have a ratio Ba:Sr:Mg;W=1:1:1:1; the usual substrates cleaning, the deposition process and postdeposition annealing.”

Point 2: Which class of technology with the new materials are applicable?

Response 2: In Introduction we inserted: “There are several technologies used for the synthesis of double perovskite structures as particles, bulk, or thin films (solid-state reaction, coprecipitation, pulsed laser deposition, rf sputtering, etc.) [16-24]”.

Point 3: This is great work. What are the technology authors are intended to mention as a future materials?

Responce 3: As we mentioned in the text the technology we proposed is spin-coating technology, suited for thin films and ultra-thin films with applications in solar cell devices, sensors photocatalysis and optoelectronics. The technology is mentioned in Experimental Section

Point 4: What are the advantages to deposit the BaSrMnWO6 with spin coating?

Response 4: The advatages of BaSrMgWO6 thin films using spin coating consist in controlling the thin film chemical composition and thickness and is a low cost technology. This is also mentioned in the response 1 and in text.